# Technical note: Uncertainty in multi-source partitioning using large tracer data sets

Alicia Correa[1,2], Diego Ochoa-Tocachi[3], Christian Birkel[1,4]

[1] Department of Geography and Water and Global Change Observatory, University of Costa Rica, San José, 2060, Costa Rica

[2] Institute for Applied Sustainability Research (iSUR), Quito, 170503, Ecuador

[3] Department of Mathematics, Universidad San Francisco de Quito, Quito, 170901, Ecuador

[4] Northern Rivers Institute, University of Aberdeen, Aberdeen, AB24 3UF, UK

*Correspondence to*: Alicia Correa (alicia.correa@ucr.ac.cr)

**Abstract**

The availability of large tracer data sets opened up the opportunity to investigate multiple source contributions to a mixture. However, the source contributions may be uncertain and apart from Bayesian approaches to estimate such source uncertainty only exist sound methods for two and three sources. We expand these methods developing an uncertainty estimation method for four sources based on multiple tracers as input data. Taylor series approximation is used to solve the set of linear mass balance equations. We illustrate the method to compute individual uncertainties in the calculation of source contributions to a mixture, particularly with an example from hydrology, where a 14-tracer set from water sources and streamflow from a tropical, high-elevation catchment were used. Moreover, this method has the potential to be generalized to any number of tracers across a range of disciplines.

## 1. Introduction

Tracer applications have dramatically increased over recent years across a wide range of disciplines (West et al., 2010). Applications in hydrology (Hooper, 2003; James and Roulet, 2006; Kirchner and Neal, 2013), ecology (Phillips and Gregg, 2003; Semmens et al., 2009b), anthropology (Ehleringer et al., 2008), conservation biology (Bicknell et al., 2014), nutrition (Magaña-Gallegos et al., 2018), environmental and ecosystem science (Bartov et al., 2013; Granek et al., 2009), and erosion and sediment transportation (Davies et al., 2018) have been the most prominent. Such a widespread use of tracers was mainly facilitated by the availability of analytical techniques that provide high sensitive, rapid multi-element analysis at lower cost (Falkner et al., 1995). For example, the use of inductively coupled plasma mass spectrometry (ICP-MS) as one of the leading analytical techniques for elemental analysis (Helaluddin et al., 2016), led to the availability and use of large tracers sets (elements) in hydrological studies (Barthold et al., 2017; Belli et al., 2017; Correa et al., 2017; Kirchner and Neal, 2013; Mimba et al., 2017). Trace elements together with water stable isotopes (Cavity Ringdown Laser Absorption Spectroscopy paved the way: (Berman et al., 2009; Lis et al., 2008)) as well as physical-chemical water parameters (e.g. electrical conductivity and pH) are now often used to improve understanding of hydro-geochemical cycles, flow pathways and runoff generation in hydrology. Furthermore, mixing models based on mass balance equations are widely-applied to identify the dominant sources and their dynamics as components of a mixture.

In hydrological mixing models the composition of the stream is assumed to be an integrated mixture of signatures of different sources (Christophersen et al., 1990). The proportional contributions of n+1 sources to the stream can be uniquely determined using n different tracers (Christophersen & Hooper, 1992). Bayesian methods have been developed to identify multiple (> 3) sources and compute their contributions to a mixture in a two-dimensional space (Parnell et al., 2010; Stock et al., 2018). In this case a unique solution is not feasible and a higher uncertainty is attributed to the model (Phillips and Gregg, 2001, 2003). On the other hand, End Member Mixing Analysis (EMMA) (Hooper, 2003) was developed to use multiple tracers as input, and therefore, allows for a multi-dimensional space that potentially increases the number of identifiable sources (Barthold et al., 2011; Inamdar et al., 2013; Liu et al., 2004). Additionally, the use of multiple tracers can avoid bias and subjectivity in the input information. Therefore, EMMA provides a robust and complete conceptualization of catchment functioning and source interactions during runoff generation (Iwasaki et al., 2015). However, despite its benefits, the EMMA approach lacks a formal methodology to assess the uncertainty of multiple end-members (Delsman et al., 2013) and to assess individual uncertainties in the calculation of source contributions to a stream.

To our knowledge, the uncertainty estimation of source contributions to streams is based on Gaussian error propagation (Genereux, 1998) and was so far only calculated using one or two tracers simultaneously (MixSIAR: Parnell et al., 2010; Phillips & Gregg, 2001; Semmens, Moore, et al., 2009). Alternatively, when the number of sources is higher, the uncertainty is usually based on the sum of analytical errors, elevation effects and the spatial variability of end-member concentrations (Uhlenbrook and Hoeg, 2003). Hence, we propose a novel and robust methodology to estimate the uncertainty of individual end-member (source) contributions to streams (mixture) based on a multi-tracer set in a three-dimensional space defined by a Principal Component Analysis.

We illustrate this application using data from Correa et al. (2019b), where the authors calculated the uncertainties only based on the application of a final equation without disclosing any details in the calculation and methodology used. The main objective of this Technical Note is therefore to explicitly describe the mathematical development

in all detail that allows the calculation of partial derivatives, degrees of freedom and confidence interval limits for

each source fraction contribution as well as to provide the code and example data for their calculation and reproducibility.

## 2.  Uncertainty estimation method development

In this section, the uncertainty estimation method presented in Phillips and Gregg, (2001) is expanded for four source contributions to the mixture.

Let $\mathcal{C}$, represents the set of sources: A, B, C and D, and mixture M, $\mathcal{C} = \{A, B, C, D, M\}$. In the following equations,

$x \in \mathcal{C}$, $y \in \{\overline{\delta}, \overline{\lambda}, \overline{\phi}\}$ and $z \in \{A, M, C\}$. $x$, $y$ and $z$ are variables that belong to the sets: $x$ to the set of A, B, C, D and mixture M, $y$ to the set of medians of every projected source and mixture in each principal component $\overline{\delta}, \overline{\lambda}, \overline{\phi}$ respectively of the used sub index and $z$ to the set of A, M and C.

If the system is composed of Eq. (1)

$$
\begin{cases}
\overline{\delta}_A f_A & + & \overline{\delta}_B f_B & + & \overline{\delta}_C f_C & + & \overline{\delta}_D f_D & = & \overline{\delta}_M \\
\overline{\lambda}_A f_A & + & \overline{\lambda}_B f_B & + & \overline{\lambda}_C f_C & + & \overline{\lambda}_D f_D & = & \overline{\lambda}_M \\
\overline{\phi}_A f_A & + & \overline{\phi}_B f_B & + & \overline{\phi}_C f_C & + & \overline{\phi}_D f_D & = & \overline{\phi}_M \\
f_A & + & f_B & + & f_C & + & f_D & = & 1
\end{cases}
\qquad \text{Eq.(1)}
$$

where $f_A, f_B, f_C$ and $f_D$ represent the contribution fraction of sources A, B, C and D respectively to the mixture M

and Eq. (1) has solution[1] for $f_A, f_B, f_C, f_D > 0$, they take the following form:

$$
\begin{aligned}
f_A & = \frac{(\Phi_M - \Delta_M)(\Lambda_C - \Delta_C) - (\Lambda_M - \Delta_M)(\Phi_C - \Delta_C)}{(\Phi_A - \Delta_A)(\Lambda_C - \Delta_C) - (\Lambda_A - \Delta_A)(\Phi_C - \Delta_C)} = \frac{Num}{Den} \qquad \text{Eq.(2)}\\
f_C & = \frac{(\Delta_M - \Lambda_M) - (\Delta_A - \Lambda_A) f_A}{(\Delta_C - \Lambda_C)}\\
f_B & = \Delta_M - (\Delta_C f_C + \Delta_A f_A)\\
f_D & = 1 - (f_C + f_B + f_A)
\end{aligned}
$$

where

$$
\Delta_x = \frac{\overline{\delta}_x - \overline{\delta}_D}{\overline{\delta}_B - \overline{\delta}_D}, \ \Lambda_x = \frac{\overline{\lambda}_x - \overline{\lambda}_D}{\overline{\lambda}_B - \overline{\lambda}_D}, \ \Phi_x = \frac{\overline{\phi}_x - \overline{\phi}_D}{\overline{\phi}_B - \overline{\phi}_D}. \qquad \text{Eq.(3)}
$$

The partial derivatives of Eq. (2) are given by:

---

[1] The system has a solution if the vector of mixture M is on the polyhedron generated by the vectors of sources A, B, C, D such that $\sum_x f_x = 1$.

$$\frac{\partial f_A}{\partial y_x} = \frac{1}{Den^2}\left[\left[(\Lambda_C - \Delta_C)\left(\frac{\partial \Phi_M}{\partial y_x} - \frac{\partial \Delta_M}{\partial y_x}\right) + (\Phi_M - \Delta_M)\left(\frac{\partial \Lambda_C}{\partial y_x} - \frac{\partial \Delta_C}{\partial y_x}\right)\right.\right.$$

$$-(\Phi_C - \Delta_C)\left(\frac{\partial \Lambda_M}{\partial y_x} - \frac{\partial \Delta_M}{\partial y_x}\right) - (\Lambda_M - \Delta_M)\left(\frac{\partial \Phi_C}{\partial y_x} - \frac{\partial \Delta_C}{\partial y_x}\right)\right] Den$$

$$-\left[(\Lambda_C - \Delta_C)\left(\frac{\partial \Phi_A}{\partial y_x} - \frac{\partial \Delta_A}{\partial y_x}\right) + (\Phi_A - \Delta_A)\left(\frac{\partial \Lambda_C}{\partial y_x} - \frac{\partial \Delta_C}{\partial y_x}\right)\right.$$

$$\left.\left.-(\Phi_C - \Delta_C)\left(\frac{\partial \Lambda_A}{\partial y_x} - \frac{\partial \Delta_A}{\partial y_x}\right) - (\Lambda_A - \Delta_A)\left(\frac{\partial \Phi_C}{\partial y_x} - \frac{\partial \Delta_C}{\partial y_x}\right)\right] Num\right]$$

$$\frac{\partial f_C}{\partial y_x} = \frac{1}{(\Delta_C - \Lambda_C)^2}\left[\left[\left(\frac{\partial \Delta_M}{\partial y_x} - \frac{\partial \Lambda_M}{\partial y_x}\right) - \left(\frac{\partial \Delta_A}{\partial y_x} - \frac{\partial \Lambda_A}{\partial y_x}\right)f_A - (\Delta_A - \Lambda_A)\frac{\partial f_A}{\partial y_x}\right](\Delta_C - \Lambda_C)\right.$$

$$\left.-\left(\frac{\partial \Delta_C}{\partial y_x} - \frac{\partial \Lambda_C}{\partial y_x}\right)[(\Delta_M - \Lambda_M) - (\Delta_A - \Lambda_A)f_A]\right],$$

$$\frac{\partial f_B}{\partial y_x} = \frac{\partial \Delta_M}{\partial y_x} - \frac{\partial \Delta_C}{\partial y_x}f_C - \Delta_C\frac{\partial f_C}{\partial y_x} - \frac{\partial \Delta_A}{\partial y_x}f_A - \Delta_A\frac{\partial f_A}{\partial y_x},$$

$$\frac{\partial f_D}{\partial y_x} = -\frac{\partial f_C}{\partial y_x} - \frac{\partial f_B}{\partial y_x} - \frac{\partial f_A}{\partial y_x}$$

Eq.(4)

It is trivial that

$$\frac{\partial \Delta_z}{\partial w_x} = 0, \ w \in \{\overline{\lambda}, \overline{\phi}\}; \ \frac{\partial \Lambda_z}{\partial w_x} = 0, \ w \in \{\overline{\delta}, \overline{\phi}\}; \ \frac{\partial \Phi_z}{\partial w_x} = 0, \ w \in \{\overline{\delta}, \overline{\lambda}\}.$$

Eq.(5)

where

$$\frac{\partial \Delta_z}{\partial \overline{\delta}_x} = (\overline{\delta}_B - \overline{\delta}_D)^{-1}\begin{cases} 1 & z \in \{A, C, M\} \text{ and } x = z \\ -\Delta_z & z \neq B \text{ and } x = B \\ \Delta_z - 1 & z \neq D \text{ and } x = D \\ 0 & otherwise \end{cases},$$

Eq.(6)

$$\frac{\partial \Lambda_z}{\partial \overline{\lambda}_x} = (\overline{\lambda}_B - \overline{\lambda}_D)^{-1}\begin{cases} 1 & z \in \{A, C, M\} \text{ and } x = z \\ -\Lambda_z & z \neq B \text{ and } x = B \\ \Lambda_z - 1 & z \neq D \text{ and } x = D \\ 0 & otherwise \end{cases} \text{ and}$$

Eq.(7)

$$\frac{\partial \Phi_z}{\partial \overline{\phi}_x} = (\overline{\phi}_B - \overline{\phi}_D)^{-1}\begin{cases} 1 & z \in \{A, C, M\} \text{ and } x = z \\ -\Phi_z & z \neq B \text{ and } x = B \\ \Phi_z - 1 & z \neq D \text{ and } x = D \\ 0 & otherwise \end{cases}.$$

Eq.(8)

For example, for $f_A$ we have

$$
\begin{aligned}
\frac{\partial f_A}{\partial \overline{\delta}_x} &= \frac{1}{Den^2}\left[\left[\frac{\partial \Delta_M}{\partial \overline{\delta}_x}(\Phi_C - \Lambda_C) - \frac{\partial \Delta_C}{\partial \overline{\delta}_x}(\Phi_M - \Lambda_M)\right]Den\right.\\
&\qquad \left. - \left[\frac{\partial \Delta_A}{\partial \overline{\delta}_x}(\Phi_C - \Lambda_C) - \frac{\partial \Delta_C}{\partial \overline{\delta}_x}(\Phi_A - \Lambda_A)\right]Num\right].\\
\frac{\partial f_A}{\partial \overline{\lambda}_x} &= \frac{1}{Den^2}\left[\left[\frac{\partial \Lambda_C}{\partial \overline{\lambda}_x}(\Phi_M - \Delta_M) - \frac{\partial \Lambda_M}{\partial \overline{\lambda}_x}(\Phi_C - \Delta_C)\right]Den\right.\\
&\qquad \left. - \left[\frac{\partial \Lambda_C}{\partial \overline{\lambda}_x}(\Phi_A - \Delta_A) - \frac{\partial \Lambda_A}{\partial \overline{\lambda}_x}(\Phi_C - \Delta_C)\right]Num\right].\\
\frac{\partial f_A}{\partial \overline{\phi}_x} &= \frac{1}{Den^2}\left[\left[\frac{\partial \Phi_M}{\partial \overline{\phi}_x}(\Lambda_C - \Delta_C) - \frac{\partial \Phi_C}{\partial \overline{\phi}_x}(\Lambda_M - \Delta_M)\right]Den\right.\\
&\qquad \left. - \left[\frac{\partial \Phi_A}{\partial \overline{\phi}_x}(\Lambda_C - \Delta_C) - \frac{\partial \Phi_C}{\partial \overline{\phi}_x}(\Lambda_A - \Delta_A)\right]Num\right].
\end{aligned}
$$

Eq.(9)

Using Eq. (9), the first-order Taylor series approximation (Taylor, 1982) for the variance of $f_A$ evaluated at the mean can be calculated by:

$$
\sigma_{f_A}^2 = \sum_x \left(\frac{\partial f_A}{\partial \overline{\delta}_x}\right)^2 \sigma_{\delta_x}^2 + \sum_x \left(\frac{\partial f_A}{\partial \overline{\lambda}_x}\right)^2 \sigma_{\lambda_x}^2 + \sum_x \left(\frac{\partial f_A}{\partial \overline{\phi}_x}\right)^2 \sigma_{\phi_x}^2 = \sum_y \sum_x \left(\frac{\partial f_A}{\partial y_x}\right)^2 \sigma_{y_x}^2.
$$

Eq.(10)

To calculate $\gamma_A$ (the Satterthwaite (1946) approximation for the degrees of freedom), we define $f_{Ay_x} = c_A \left(\frac{\partial f_A}{\partial y_x}\right)^2$. In this case, we get:

$$
\gamma_A = \frac{\left(\sum_y \sum_x f_{Ay_x} \sigma_{y_x}^2\right)^2}{\sum_y \sum_x \frac{\left(f_{Ay_x}\sigma_{y_x}^2\right)^2}{n_{y_x} - 1}}.
$$

Eq.(11)

Note that whatever the value of $c_A$ is, Eq. (11) leads to:

$$
\gamma_A = \frac{\left(\sum_y \sum_x \left(\frac{\partial f_A}{\partial y_x}\right)^2 \sigma_{y_x}^2\right)^2}{\sum_y \sum_x \frac{\left(\left(\frac{\partial f_A}{\partial y_x}\right)^2 \sigma_{y_x}^2\right)^2}{n_{y_x} - 1}}
$$

and if we set $f_{Ay_x}^* = \left(\frac{\partial f_A}{\partial y_x}\right)^2$ then the numerator of the last equation can be replaced by $\left(\sigma_{f_A}^2\right)^2$. In other words, we can use Eq. (10) and the derivatives (9) to estimate the value of $\gamma_A$ resulting in $f_{Ay_x} = c_A f_{Ay_x}^*$. Of course, it is required that $c_A$ is constant w.r.t. $y_x$. Then,

$$\gamma_A = \frac{\left(\sigma_{f_A}^2\right)^2}{\sum_y \sum_x \dfrac{\left(\left(\frac{\partial f_A}{\partial y_x}\right)^2 \sigma_{y_x}^2\right)^2}{n_{y_x} - 1}} \qquad \text{Eq.(12)}$$

Let $w \in \mathcal{C} \setminus \{A\}$. The first-order Taylor series approximation for the variance of $f_w$, can be calculated by (as above):

$$\sigma_{f_w}^2 = \sum_x \left(\frac{\partial f_w}{\partial \overline{\delta_x}}\right)^2 \sigma_{\overline{\delta_x}}^2 + \sum_x \left(\frac{\partial f_w}{\partial \overline{\lambda_x}}\right)^2 \sigma_{\overline{\lambda_x}}^2 + \sum_x \left(\frac{\partial f_w}{\partial \overline{\phi_x}}\right)^2 \sigma_{\overline{\phi_x}}^2 = \sum_y \sum_x \left(\frac{\partial f_w}{\partial y_x}\right)^2 \sigma_{y_x}^2. \qquad \text{Eq.(13)}$$

If we construct $\gamma_w$ as $\gamma_A$, we get:

$$\gamma_w = \frac{\left(\sum_y \sum_x f_{wy_x}^* \sigma_{y_x}^2\right)^2}{\sum_y \sum_x \dfrac{\left(f_{wy_x}^* \sigma_{y_x}^2\right)^2}{n_{y_x} - 1}}$$

where $f_{wy_x} = c_w f_{wy_x}^*$ and $f_{wy_x}^* = \left(\frac{\partial f_w}{\partial y_x}\right)^2$ with $c_w$ constant w.r.t. $y_x$, then we finally get:

$$\gamma_w = \frac{\left(\sigma_{f_w}^2\right)^2}{\sum_y \sum_x \dfrac{\left(\left(\frac{\partial f_w}{\partial y_x}\right)^2 \sigma_{y_x}^2\right)^2}{n_{y_x} - 1}} \qquad \text{Eq.(14)}$$

The upper and lower confidence interval limits for each end-member fraction can be calculated using partial derivatives and the 95% confidence intervals constructed as:

$$f_w \pm t_{0.05\gamma_w} \sigma_{fw} \qquad \text{Eq.(15)}$$

Where $t_{0.05,\gamma}$ is the Student's t for $\alpha=0.05$ (two-tailed) (Walpole et al., 2017) and $\gamma$ degrees of freedom related with $\sigma_{fw}$.

### 3. Application

#### 3.1. Study site and data

This methodology was tested using data from a high elevation (3,500 - 3,900 m a.s.l.) tropical catchment (7.53 km²) located in southern Ecuador (3°4′38″S, 79°15′30″O). The mean annual precipitation for this study site is 1,300 mm (Padrón et al., 2015), the mean annual discharge is 860 mm yr⁻¹. The catchment is of a volcanic origin and dominated by volcanic Histosol (24%) and Andosol (72%) soils (IUSS Working Group WRB, 2015), both with high percentage of organic matter content (450 and 310 g kg, respectively) (Quichimbo et al., 2012) and large water-holding capacities (Buytaert et al., 2006). Histosols are primarily located at the valleys and covered by cushion plants, while Andosol soils are predominated on the hillslopes under a cover of tussock grass. Nearly-saturated conditions of the riparian zone are observed year-round, and a spring is located in the north-western part of the catchment. Streamwater samples from 5 nested streams were collected weekly from March 2013 to April 2014 (n=257) and at a higher frequency during experimental campaigns. Additionally, bi-weekly water samples from 4 potential end-members: rainfall (RF), soil water from Andosols (AN) and Histosols (HS) and spring water (SW) (n ~ 30, for each end-member) were collected. The above-mentioned waters sources (RF, AN, HS and SW), were previously identified as end-members (Correa et al., 2017, 2019b) (Table 1). A multi-tracer (14 tracers) data set of conservative tracers was obtained from each water sample (Na, Mg, Al, Si, K, Ca, Rb, Sr, Ba, Ce, V, Y, Nd) at the Institute for Landscape Ecology and Resource Management of the Justus Liebig University using an ICP-MS (Agilent 7500ce, Agilent Technologies) and the electrical conductivity (EC) was measured in situ. More detailed information on the study site and data set can be found in Correa et al., (2017, 2019b).

#### 3.2. Uncertainty estimation of water source contributions

Using the classic EMMA approach (Christophersen and Hooper, 1992), end-members (source) and stream (mixture) data were projected into a three-dimensional space (Correa et al., 2019b) visualized in Figure 1. The resulting median and standard deviation of end-members and stream coordinates are shown in Table 1. Furthermore, Figure 2 shows the distribution of projected samples from individual end-members in the PCA coordinates.

The uncertainty of each of the four end-member contributions to the stream was determined using the above developed first-order Taylor series approximation from Eq. 14 (MatLab code in (Correa et al., 2019a). The variance for each end-member fraction was calculated (as recommended by Phillips and Gregg (2001)) as $f_{EM1} \pm t_{0.05,\gamma} \sigma_{fEM1}$. The $t_{0.05,\gamma}$ depicts the Student's t for $\alpha$=0.05 (two-tailed) and $\gamma$ degrees of freedom. The $\gamma$ degrees of freedom represents the Satterthwaite (1946) approximation for the related degrees of freedom with $\sigma_{fEM1}$ and can be calculated as follows:

$$\gamma_{EM1} = \frac{\left(\sigma_{fEM1}^2\right)^2}{\sum_y \sum_x \dfrac{\left(\left(\dfrac{\partial f_{EM1}}{\partial y_x}\right)^2 \sigma_{y_x}^2\right)^2}{n_{y_x} - 1}} \qquad \text{Eq.(16)}$$

Note that Eq. (16) is an adaptation of Eq. (14) for this particular end-member configuration with x = EM1,
       EM3, EM4 and M, y = $\overline{\delta}, \overline{\lambda}$ and $\overline{\phi}$, n= number of samples. The $\overline{\delta}, \overline{\lambda}$ and $\overline{\phi}$, represent the median of the projected
       water samples from end-members and stream in the principal components U1, U2 and U3, respectively (U1
       represents the principal components PC1, U2 PC2 and U3 and PC3). The $f_{EM1}$ gives w the proportion of EM1 in
       M and $\sigma^2_{f_{EM1}}$, the variances of the EM1. A similar procedure should be used for all end-members. The resulting
uncertainty estimates for each source end-member are shown in Table 5.

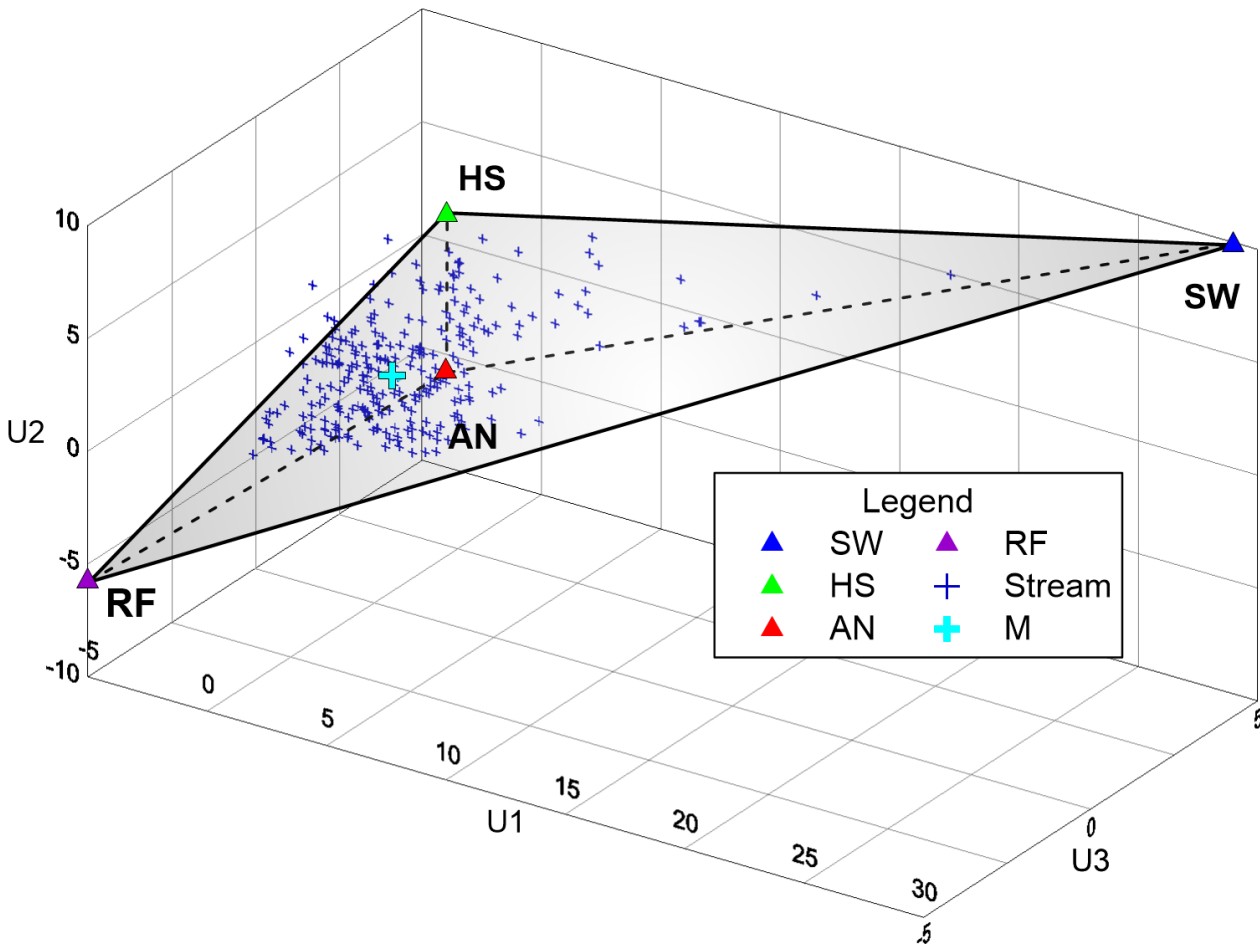

**Figure 1. Three-dimensional mixing space generated using stream data, where the median of end-members are projected. U1 represents 59.6% of the variance, U2 19.7%, and U3 7.4% (From PCA); RF, rainfall; AN, Andosols; HS, Histosols; SW, spring water; M, median of stream data (mixture)**

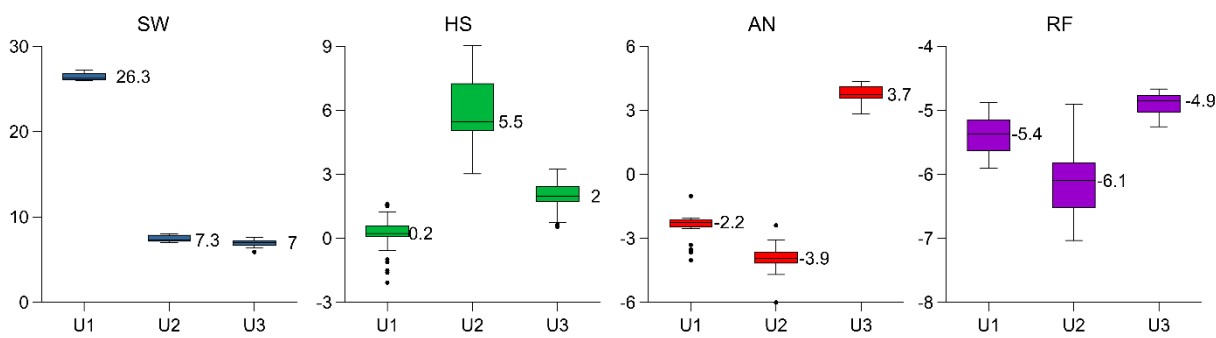

**Figure 2. Boxplots of end-members projected in the three-dimensional mixing space for the study period 2013–2014, the Y-axis represents the coordinates of the mixing space and the X-axis the principal components U1, U2 and U3 (the**

**central bar in the box represents the median; notches represent the 95% confidence intervals; whiskers 1.5 times the interquartile range and circles represent outliers). SW, spring water; HS, Histosol; AN, Andosol; RF, rainfall.**

From the above-mentioned data set, we have generated 6 examples to assess the sensitivity of the uncertainty calculation to the source sample size, the artificial inclusion of outliers (upper and lower extremes) and the increased standard deviations of the source datasets.

The first example considers 50% of the samples from each source. The median, standard deviation and sample size are input data (Table 2) to calculate the uncertainty ranges (Table 6).

- The first example considers 50% of the samples from each source. The median, standard deviation and sample size are input data (Table 2) to calculate the uncertainty bands (Table 6).

- The second considers the remaining 50% of samples and was similarly executed (Table 2).

- In the third example, outliers were artificially included at the upper positive end of data sets for each source at each coordinate, respectively. The outliers consisted of twice the maximum positive value of the observed data (Table 3).

- Using the same criteria, the negative extremes were included in the fourth example (Table 3).

- Sources affected by dispersed data clouds were taken into account by an increase in the standard deviation. We considered two cases, the first, in the example five, increasing three times the value of the standard deviation of the initial data set (Table 4) and finally, increasing the standard deviation five times for the sixth example (Table 4).

The results of this analysis are presented in Tables 6-8. In examples 1 and 2 the sample size reduction from 24 to 12 and 13 samples respectively (Table 6), had a minimal effect (less than 3%) on the calculation of the uncertainty ranges compared to the original complete set (Table 1). The fractions of source contributions did not experience changes. The inclusion of outliers affected the values of the medians at levels of the second decimal (Table 3) in relation to the median of the initial data (Table 2). However, the standard deviations increased in a range of 1.2 to 2.5 times the original value for AN and HS, and more for RF (2.5 to 10.5) and drastically for SW (4 to 20 times wider). These variations were reflected in the results of the calculation of uncertainties where the limits were extended for all existing cases from 1% to 12% (Table 6) in relation to Table 5. Furthermore, the widening of the standard deviations to three and five times their initial values resulted in an increase in the range of uncertainty between 2% and 22% for the first case and between 5% and 37% for the second case. For the latter, the minimum limit of the uncertainty range was reached in all the reported cases. The large number of samples used in these exercises were reflected in high degrees of freedom.

## 4.  Summary and remarks

Our methodology developed to calculate the contribution of sources to the mixture and its associated uncertainty (based on multiple tracer sets) has been shown to be effective in real application cases. The robustness of the method is reflected in the fact that the calculations of the uncertainty ranges of multiple source contributions to a mixture do not experience significant changes with sample size reduction or inclusion of outliers. Rather, it shows marginally different results by incorporating standard deviations from widely dispersed data.

The methodology, based on Phillips and Gregg, (2001) combined with EMMA applications (Hooper, 2003) presents high potential for use as an alternative method to the simple sum of analytical errors (Uhlenbrook and Hoeg, 2003) or the Bayesian approach (Parnell et al., 2010; Stock et al., 2018). We provide a tool to help the community that has reported that a greater number of sources contribution and (common 2 or 3) the related uncertainty is needed for a more complete conceptualization of the mixing processes (Iwasaki et al., 2015).

The MatLab code provided and the illustrative examples facilitate the understanding of the methodology and promote future scientific applications. We are confident that the use of this methodology will help the scientific community that is increasingly using large tracer sets in its research to obtain robust results.

## 5.  Code and data availability

A MatLab code to calculate the fractions of end-members contribution to the mixture and their associated uncertainties is freely available in *https://zenodo.org/record/2649201*. As well as input data (used in this study) as an example for the code run and an instruction note.

## 6.  Author contribution

AC and CB conceptualized the methodology. AC was responsible for the data collection and analysis. DO AC programmed and evaluated the MatLab code with collected data. AC wrote the manuscript with contributions from all co-authors.

## 7.  Competing interests

The authors declare that they have no conflict of interest

**Acknowledgements**

AC and CB would like to acknowledge support by a UCR postdoctoral fellowship awarded to AC, financial support by UCREA awarded to CB and the Water and Global Change Observatory at the Department of Geography, UCR. The authors thank the Central Research Office (DIUC) of the Universidad de Cuenca for making available part of the tracer data sets. We are especially grateful for the constructive comments that were provided by the referees, which greatly improved the quality of the Technical Note.

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

**Table 1. Median and standard deviation (std.dev.) of end-members and stream projected in three-dimensional space for the study period 2013–2014.**

| End-member | | Coordinates* | | | Naming |
|---|---|---|---|---|---|
| | | U1 | U2 | U3 | in equations |
| SW (n = 25) | median | 26,25 | 7,29 | 7,00 | A |
| | std.dev. | 0,46 | 0,36 | 0,39 | |
| HS (n = 33) | median | 0,23 | 5,48 | 1,97 | B |
| | std.dev. | 0,85 | 1,29 | 0,69 | |
| AN (n = 37) | median | -2,24 | -3,93 | 3,71 | C |
| | std.dev. | 0,55 | 0,58 | 0,45 | |
| RF (n = 36) | median | -5,38 | -6,10 | -4,84 | D |
| | std.dev. | 0,27 | 0,56 | 0,15 | |
| Stream (n = 257) | median | -0,61 | -1,04 | 0,94 | M |
| | std.dev. | 2,06 | 1,10 | 0,66 | |

* Coordinates of end-members and stream (mixture) medians in three-dimensional space (U1, U2 and U3). n represents the sample size.

**Table 2. Median and standard deviation (std.dev.) of end-members and stream projected in three-dimensional considering 50% of the data sets**

| Naming in equations | | 1) | End member | Coordinates* | | | 2) | End member | Coordinates* | | |
|---|---|---|---|---|---|---|---|---|---|---|---|
| | | | | U1 | U2 | U3 | | | U1 | U2 | U3 |
| A | median | | SW | 26.18 | 7.29 | 6.66 | | SW | 26.28 | 7.29 | 7.1 |
| | std.dev. | | (n = 12) | 0.34 | 0.39 | 0.48 | | (n = 13) | 0.51 | 0.36 | 0.21 |
| B | median | | HS | 0.23 | 5.41 | 1.87 | | HS | 0.28 | 5.9 | 2.26 |
| | std.dev. | | (n = 17) | 0.74 | 1.19 | 0.52 | | (n = 17) | 0.96 | 1.33 | 0.74 |
| C | median | | AN | -2.37 | -3.93 | 3.69 | | AN | -2.2 | -3.94 | 3.89 |
| | std.dev. | | (n = 19) | 0.59 | 0.4 | 0.49 | | (n = 19) | 0.46 | 0.73 | 0.41 |
| D | median | | RF | -5.37 | -6.26 | -4.78 | | RF | -5.35 | -5.99 | -5.01 |
| | std.dev. | | (n = 18) | 0.26 | 0.58 | 0.07 | | (n = 18) | 0.28 | 0.53 | 0.15 |
| M | median | | Stream | -0,61 | -1,04 | 0,94 | | Stream | -0,61 | -1,04 | 0,94 |
| | std.dev. | | (n = 257) | 2,06 | 1,10 | 0,66 | | (n = 257) | 2,06 | 1,10 | 0,66 |

The example 1) considers the initial 50% and 2) the remaining 50% of the sample sets.* Coordinates of end-members and stream (mixture) medians in three-dimensional space (U1, U2 and U3). n represents the sample size.

**Table 3. Median and standard deviation (std.dev.) of end-members and stream projected in three-dimensional including artificial outliers**

| Naming in equations | | 3) | End member | Coordinates* | | | 4) | End member | Coordinates* | | |
|---|---|---|---|---|---|---|---|---|---|---|---|
| | | | | U1 | U2 | U3 | | | U1 | U2 | U3 |
| A | median | | SW | 26.25 | 7.3 | 7.02 | | SW | 26.21 | 7.29 | 6.95 |
| | std.dev. | | (n = 26) | 5.51 | 1.73 | 1.68 | | (n = 26) | 10.28 | 2.87 | 2.54 |
| B | median | | HS | 0.27 | 5.47 | 1.98 | | HS | 0.23 | 5.45 | 1.97 |
| | std.dev. | | (n = 34) | 0.99 | 2.45 | 1.03 | | (n = 34) | 1.12 | 1.99 | 0.8 |
| C | median | | AN | -2.24 | -3.92 | 3.79 | | AN | -2.26 | -3.95 | 3.74 |
| | std.dev. | | (n = 38) | 0.78 | 1.17 | 0.92 | | (n = 38) | 1.07 | 1.43 | 1.15 |
| D | median | | RF | -5.36 | -6.08 | -4.84 | | RF | -5.37 | -6.11 | -4.86 |
| | std.dev. | | (n = 37) | 1.7 | 1.89 | 1.58 | | (n = 37) | 1.09 | 1.42 | 0.94 |
| M | median | | Stream | -0,61 | -1,04 | 0,94 | | Stream | -0,61 | -1,04 | 0,94 |
| | std.dev. | | (n = 257) | 2,06 | 1,10 | 0,66 | | (n = 257) | 2,06 | 1,10 | 0,66 |

The example 3) considers outliers included at the positive extreme of the dataset of each source and 4) outliers included at the negative extreme.* Coordinates of end-members and stream (mixture) medians in three-dimensional space (U1, U2 and U3). n represents the sample size.

**Table 4. Median and enlarged standard deviation (std.dev.) of end-members and stream projected in three-dimensional**

| Naming in equations | | 5) | End member | Coordinates* | | | 6) | End member | Coordinates* | | |
|---|---|---|---|---|---|---|---|---|---|---|---|
| | | | | U1 | U2 | U3 | | | U1 | U2 | U3 |
| A | median | | SW | 26,25 | 7,29 | 7,00 | | SW | 26,25 | 7,29 | 7,00 |
| | std.dev. | | (n = 25) | 1.39 | 1.07 | 1.19 | | (n = 25) | 2.32 | 1.78 | 1.99 |
| B | median | | HS | 0,23 | 5,48 | 1,97 | | HS | 0,23 | 5,48 | 1,97 |
| | std.dev. | | (n = 33) | 2.56 | 3.87 | 2.06 | | (n = 33) | 4.27 | 6.45 | 3.43 |
| C | median | | AN | -2,24 | -3,93 | 3,71 | | AN | -2,24 | -3,93 | 3,71 |
| | std.dev. | | (n = 37) | 1.65 | 1.73 | 1.34 | | (n = 37) | 2.75 | 2.88 | 2.24 |
| D | median | | RF | -5,38 | -6,10 | -4,84 | | RF | -5,38 | -6,10 | -4,84 |
| | std.dev. | | (n = 36) | 0.8 | 1.69 | 0.46 | | (n = 36) | 1.34 | 2.81 | 0.77 |
| M | median | | Stream | -0,61 | -1,04 | 0,94 | | Stream | -0,61 | -1,04 | 0,94 |
| | std.dev. | | (n = 257) | 2,06 | 1,10 | 0,66 | | (n = 257) | 2,06 | 1,10 | 0,66 |

The example 5) considers 3-times the standard deviation of the original data set and 6) 5-times the standard deviation of the original data set.* Coordinates of end-members and stream (mixture) medians in three-dimensional space (U1, U2 and U3). n represents the sample size.

**Table 5. Uncertainty of individual end-member contributions to the stream and Satterthwaite (1946) approximation for the degrees of freedom calculated for the study period 2013–2014**

|  | EM1 SW | EM2 HS | EM3 AN | EM4 RF |
|---|---|---|---|---|
| Fraction of end-members contribution | 0.06 | 0.3 | 0.35 | 0.29 |
| Upper 95% confidence limit | 0.21 | 0.57 | 0.58 | 0.46 |
| Lower 95% confidence limit | 0.00 | 0.03 | 0.12 | 0.12 |
| Degrees of freedom | 291 | 536 | 749 | 628 |

**Table 6. Uncertainty of individual end-member contributions to the stream and Satterthwaite (1946) approximation for the degrees of freedom computed considering 50% of the data sets**

|  | 1) | EM1 SW | EM2 HS | EM3 AN | EM4 RF | 2) | EM1 SW | EM2 HS | EM3 AN | EM4 RF |
|---|---|---|---|---|---|---|---|---|---|---|
| Fraction of end-members contribution |  | 0.06 | 0.3 | 0.35 | 0.28 |  | 0.06 | 0.28 | 0.35 | 0.3 |
| Upper 95% confidence limit |  | 0.21 | 0.57 | 0.58 | 0.45 |  | 0.21 | 0.55 | 0.58 | 0.46 |
| Lower 95% confidence limit |  | 0.00 | 0.03 | 0.12 | 0.11 |  | 0.00 | 0.02 | 0.12 | 0.14 |
| Degrees of freedom |  | 289 | 493 | 676 | 589 |  | 288 | 491 | 679 | 537 |

The example 1) was computed considering the initial 50% and 2) the remaining 50% of the sample sets.

**Table 7. Uncertainty of individual end-member contributions to the stream and Satterthwaite (1946) approximation for the degrees of freedom computed after including artificial outliers**

|  | 3) | EM1 SW | EM2 HS | EM3 AN | EM4 RF | 4) | EM1 SW | EM2 HS | EM3 AN | EM4 RF |
|---|---|---|---|---|---|---|---|---|---|---|
| Fraction of end-members contribution |  | 0.06 | 0.3 | 0.35 | 0.29 |  | 0.06 | 0.3 | 0.35 | 0.29 |
| Upper 95% confidence limit |  | 0.22 | 0.62 | 0.64 | 0.5 |  | 0.22 | 0.61 | 0.63 | 0.49 |
| Lower 95% confidence limit |  | 0.00 | 0.00 | 0.06 | 0.08 |  | 0.00 | 0.00 | 0.07 | 0.08 |
| Degrees of freedom |  | 350 | 448 | 640 | 529 |  | 353 | 554 | 757 | 621 |

The example 3) was computed after including outliers at the positive extreme of the dataset and 4) including outliers at the negative extreme.

**Table 8. Uncertainty of individual end-member contributions to the stream and Satterthwaite (1946) approximation for the degrees of freedom computed with enlarged standard deviations**

|  | 5) | EM1 SW | EM2 HS | EM3 AN | EM4 RF | 6) | EM1 SW | EM2 HS | EM3 AN | EM4 RF |
|---|---|---|---|---|---|---|---|---|---|---|
| Fraction of end-members contribution |  | 0.06 | 0.3 | 0.35 | 0.29 |  | 0.06 | 0.3 | 0.35 | 0.29 |
| Upper 95% confidence limit |  | 0.23 | 0.68 | 0.69 | 0.52 |  | 0.26 | 0.83 | 0.83 | 0.61 |
| Lower 95% confidence limit |  | 0.00 | 0.00 | 0.01 | 0.05 |  | 0.00 | 0.00 | 0.00 | 0.00 |
| Degrees of freedom |  | 372 | 225 | 362 | 312 |  | 335 | 122 | 211 | 172 |

The example 5) was computed considering 3-times the standard deviation of the original data set and 6) 5-times the standard deviation of the original data set.