# Peer review of "Technical note: Uncertainty in multi-source partitioning using large tracer data sets"

_Hydrology and Earth System Sciences, 2019_

## Short Comment (SC1) · 28 May 2019

I read this technical note with interest. However, in my view, this technical note is too condensed in its form. The reader has to guess his/her way through parts of the methods and the results section is extremely short, with a single example and no discussion of basic aspects of source attribution (e.g. effect of sample size). My comments hereafter are meant to increase the readability of the technical note. I think it would be great for the readers to have some more results for the presented case study.

Some detailed comments:

- the methods section does not say what the sets C, y and z are, nor what f is. The reader can deduce it after reading the different equations but this form of presenting

the notation is unusual in the geosciences literature.

-the estimation of the confidence interval is not presented in the methods section, only the estimation of the degree of freedom. The actual confidence interval comes in the results section; even if we all know Student's law, why not include it here?

- the results section could explicitely say what U1, U2, U3 is. I only understood after going back to the introduction and combining that information with the one from the figure caption.

- would be nice to have some illustrations of how the method reacts e.g. to outliers in the source samples ? I was for example surprised to see a relatively narrow confidence interval for end-member HS, which has a high standard deviation (Table 2). The computed degree of freedoms are missing.

---

## Author Comment (AC1) · 21 Jun 2019

We are very grateful to Bettina Schaefli for her thoughtful comments, which we will fully consider in order to improve our Technical Note. We appreciate the general comment that the Technical Note is very condensed as our intentions were to emphasize the mathematical development for potential applications in future case studies (Matlab code for reproducibility was supplied). However, we fully agree that we should extend on the explanations for clarification and readability, particularly in the results and discussion sections.

Regarding the more detailed comments:

1) f represents the fraction of contributions of sources A, B, C and D, respectively of

the used sub-index. C represents the set of sources A, B, C, D and mixture M x, y and z are variables that belong to the sets: x to the set of A, B, C, D and mixture M y to the set of standard deviations in each principal component z to the set of ðİŘť, ðİŚĂ and C x, y and z are used as aids used to make the mathematical description of the equations more compressed and understandable. This will be further described in the methods section, but we prefer to maintain standard mathematical formulations since we focus our efforts on the mathematical development of the three-dimensional application (not previously developed).

2) We will include the equation and description of the estimation of the confidence interval accompanied by a reference to the t student table in the revised manuscript.

3) U1 U2 U3 are widely used terms within the EMMA literature representing the principal components applied to the chemistry of the mixture, being PC1, PC2 and PC3 respectively. However, we will describe these variables in more detail in the methodology and the results sections of the revised manuscript.

4) We will include the calculated degree of freedoms in a table and intend to create a virtual example with subsets of data to show the effect of outliers on calculated uncertainty bounds in terms of a simple sensitivity test.

---

## Referee Comment (RC1) · Anonymous Referee #1 · 29 Jun 2019

The manuscript entitled "Technical note: Uncertainty in multi-source partitioning using large tracer data sets" by Correa et al. presents a method to estimate the uncertainty of mixing models in flow partitioning for four different water sources (end-members) contributing to a mixing hydrologic system. They use a Taylor series approximation to solve the set of mass balance equations and apply it to a tracer dataset from an experimental catchment in Ecuador. Even though I appreciate the authors' effort to provide the tracer community with a potentially valuable tool (code freely available) to evaluate uncertainty in multi-source mixing systems, I agree with the other reviewer of the manuscript, Bettina Schaefli (further refereed as BS), that it is too condensed in its current form. This factor makes the development of the system of equations' solution difficult to follow, and this must be improved. In addition, I am a bit concerned that

this method and the presented illustration example have been reported previously by (Correa et al., 2019). Therefore, I strongly encourage the authors to: 1) mention openly whether their uncertainty estimation method is different than/similar to (the same?) the one applied in the aforementioned paper and ii) to apply their method to 1-2 additional illustration examples from the published literature or, at the very least, include the evaluation of outliers suggested by BS to differentiate this work from the one of (Correa et al., 2019).

Major Comments:

- The authors claim the robustness of their method (P.2, L.34), but do not discuss this consideration in comparison for example, to the commonly applied Gaussian error propagation approach. I think it would be great to evaluate and discuss this in the manuscript to add value to the usefulness of the presented methodology. This could also help differentiate this work from the work of (Correa et al., 2019).

- System of equations and resolution: please make sure to define clearly all of the notation in the set of equations to facilitate the readability of their resolution throughout the paper. All the considerations within the resolution of the system must also be clearly stated.

Specific comments:

- Page 1, Lines 12-13: this sentence is incomplete. Please correct.

- P.1, L15: "dataset".

- P.2, L.7: Replace "novel" by "the availability of".

- P.2, L.12: Delete "novel".

- P.2, footnote: I think "M" refers to the mixture, not to a source. Correct if necessary.

- P.7, L.4: n is approximately 30 for each source, or among all sources

[Figure]

- P.7, L.21 and L.23: I think you refer to streamflow (or mixture, M), and not to spring water (SW). Correct if necessary.

- Table 2: report $\lambda$ values

REFERENCED LITERATURE

Correa, A., Breuer, L., Crespo, P., Célleri, R., Feyen, J., Birkel, C., Silva, C. and Windhorst, D.: Spatially distributed hydro-chemical data with temporally high-resolution is needed to adequately assess the hydrological functioning of headwater catchments, Sci. Total Environ., 651, 1613–1626, doi:10.1016/J.SCITOTENV.2018.09.189, 2019.

---

## Referee Comment (RC2) · Anonymous Referee #2 · 2 Jul 2019

The technical note presented by Correa et al. details a method to estimate contributions from different sources to a mixture. The novelty of the work presented by the authors lies in the estimation of uncertainties of the different end member contributions. In addition to deriving the equations of the methodology, the authors also present a MATLAB Code that can be used to calculate the contributions of the different end members. I think the authors present a potentially valuable tool for the hydrologic community. However, I echo concerns regarding the readability of the technical note previously made by Bettina Schaefli and an anonymous referee. More details and explanations are required for this technical note to become beneficial to the community.

Major comments:

- The main problem with this technical note is the lack of definitions and explanations

of variables and notations. Albeit I understand that a technical note is supposed to be short, if the authors want this work to be utilized by other geoscientists and hydrologists, they need to improve the readability of the technical note.

- If I understand correctly, this technical note details the methodology used by Correa et al. (2019). It would be great if the authors could be more upfront about this in the technical note.

- The Matlab script is relatively hard to follow and understand, due to insufficient comments and documentation:

Naming of Matlab Files: Matlab filenames starting with a number cannot be run by Matlab. Please rename in order to avoid confusion for unexperienced Matlab users.

Name of csv file is different in repository than as used in matlab script (5_data.csv rather than data.csv). Please make consistent.

The Matlab Script is not very well documented. Neither the script directly, nor the readme pdf explain the actual inputs (only some cryptic A/B/C/D/M without explanations; one has to refer to Table 1 to understand the setup of the file) nor the outputs. Consequently, the script cannot be used without reading the manuscript in detail.

It would furthermore be helpful to reference the code lines to the equations of the manuscript and state in the script what each command is doing.

Please also reference the manuscript in the readme.pdf and the main script.

Specific Comments:

p.1 line 13: the grammar of this sentence is wrong/the sentence is incomplete.

p.1 lines 15-17: There is no connection between the two sentences, in spite of the "however". Sentence 1 talks about large tracer sets from four water sources (but says nothing about the number of tracers), Sentence 2 says the approach can be generalized to any number of tracers. Please make this clearer.

[Figure]

p.5 line 1-2: the reference of Taylor, 1982 should probably follow directly after "Taylor series approximation".

p.7 line 2: the specification of n=270 is no useful without also specifying the number of different streams sampled.

References: Correa et al, 2018, SciTotEnv should actually be Correa et al. 2019.

Table 1: Footnote: "three- axes". I believe this should be three-dimensional space.

Table 2: Some of the values provided here do not match up with those calculated by the MATLAB Code. Please verify.

Referenced Literature: Correa, A., Breuer, L., Crespo, P., Célleri, R., Feyen, J., Birkel, C., Silva, C. and Windhorst, D.: Spatially distributed hydro-chemical data with temporally high-resolution is needed to adequately assess the hydrological functioning of headwater catchments, Science of The Total Environment, 651, 1613–1626, doi:10.1016/j.scitotenv.2018.09.189, 2019.

---

## Author Comment (AC2) · 16 Sep 2019

Response to comments from the Referee 1:

I agree with the other reviewer of the manuscript, Bettina Schaefli (further refereed as BS), that it is too condensed in its current form. This factor makes the development of the system of equations' solution difficult to follow, and this must be improved. In addition, I am a bit concerned that this method and the presented illustration example have been reported previously by (Correa et al., 2019). Therefore, I strongly encourage the authors to: 1) mention openly whether their uncertainty estimation method is different than/similar to (the same?) the one applied in the aforementioned paper and ii) to apply their method to 1-2 additional illustration examples from the published literature or, at

the very least, include the evaluation of outliers suggested by BS to differentiate this work from the one of (Correa et al., 2019).

R: We are very thankful to the Referee's useful remarks, which greatly helped to improve our Technical Note. We appreciate the comment that the Technical Note is condensed, and it should be extended and clarified to provide the community with an easy-to-follow reading material, mainly in the description of the system of equations', their development and solution. Regarding 1) We thank the Referee for highlighting that we based our application on the example published in Correa et al (2019). However, the authors calculated the uncertainties based only on the application of a final equation. The main objective of this Technical Note is, therefore, to explicitly describe the mathematical development in all detail that allows the calculation of partial derivatives, degrees of freedom and confidence interval limits for each source fraction contribution as well as to provide the code and example data for their calculation and reproducibility. Regarding 2) An evaluation of outliers as well as four additional examples from the same data set were included in the new version of the Technical Note. Additionally, a new figure (Figure. 2) was included showing Boxplots of end-members projected in the three-dimensional mixing space as a basis for clarity and a better understanding of the example calculations. Please find as supplement the description of the examples and tables with input information (Table 1 to Table 4) and results (Table 5 to Table 8) that have been included and discussed in the Technical Note.

Major Comments: The authors claim the robustness of their method (P.2, L.34), but do not discuss this consideration in comparison for example, to the commonly applied Gaussian error propagation approach. I think it would be great to evaluate and discuss this in the manuscript to add value to the usefulness of the presented methodology. This could also help differentiate this work from the work of (Correa et al., 2019).

R: We agree with the Reviewer that an exhaustive comparison of different methods should be attempted at some point, but after careful consideration, we did not follow this suggestion here due to the length of this technical note. We on purpose used

this format and not a full research paper to present our novel methodology in more mathematical detail than usual, step-by-step with an example application and we also provide the codes. This uncertainty assessment method was not presented in Correa et al. (2019), only parts of the dataset.

System of equations and resolution: please make sure to define clearly all of the notation in the set of equations to facilitate the readability of their resolution throughout the paper. All the considerations within the resolution of the system must also be clearly stated.

R: We appreciate this suggestion and have extended and updated the manuscript to clarify the notation and also to include more details in the descriptions of equations and variables to improve the readability of the technical note. Specific comments: Page 1, Lines 12-13: this sentence is incomplete. Please correct. R: We have edited the phrase. It now reads: "[. . .], Bayesian approaches to estimate such source uncertainty only exist only sound methods for two and three sources."

P.1, L15: "dataset".

R: In this context, the word "set" refers to the set of equations used to calculate the uncertainty of the source's contributions to a mixture, not to the data set, therefore we have omitted this change.

P.2, L.7: Replace "novel" by "the availability of".

R: This has been corrected to "the availability of"

P.2, L.12: Delete "novel".

R: The word "novel" has been deleted.

P.2, footnote: I think "M" refers to the mixture, not to a source. Correct if necessary.

R: This has been corrected to "mixture"

P.7, L.4: n is approximately 30 for each source, or among all sources

R: We have edited the phrase to clarify this point. It now reads: "[. . .] and spring water (SW) (n $\sim$ 30, for each end-member) were collected".

P.7, L.21 and L.23: I think you refer to streamflow (or mixture, M), and not to spring water (SW). Correct if necessary.

R: We have edited the phrase to correct this error. "SW" was replaced by "M".

Table 2: report $\lambda$ values R: By $\lambda$ we assumed that the Referee refers to degrees of freedom ($\gamma$). These values are reported in Table 5.

Please find our response letter in the attached file.

Please also note the supplement to this comment:
https://www.hydrol-earth-syst-sci-discuss.net/hess-2019-197/hess-2019-197-AC2-supplement.pdf

**Supplement:**

**Response to comments from the Referee 1:**

**I agree with the other reviewer of the manuscript, Bettina Schaefli (further refereed as BS), that it is too condensed in its current form. This factor makes the development of the system of equations' solution difficult to follow, and this must be improved. In addition, I am a bit concerned that this method and the presented illustration example have been reported previously by (Correa et al., 2019). Therefore, I strongly encourage the authors to: 1) mention openly whether their uncertainty estimation method is different than/similar to (the same?) the one applied in the aforementioned paper and ii) to apply their method to 1-2 additional illustration examples from the published literature or, at the very least, include the evaluation of outliers suggested by BS to differentiate this work from the one of (Correa et al., 2019).**

R: We are very thankful to the Referee's useful remarks, which greatly helped to improve our Technical Note. We appreciate the comment that the Technical Note is condensed, and it should be extended and clarified to provide the community with an easy-to-follow reading material, mainly in the description of the system of equations', their development and solution.

Regarding 1) We thank the Referee for highlighting that we based our application on the example published in Correa et al (2019). However, the authors calculated the uncertainties based only on the application of a final equation. The main objective of this Technical Note is therefore to explicitly describe the mathematical development in all detail that allows the calculation of partial derivatives, degrees of freedom and confidence interval limits for each source fraction contribution as well as to provide the code and example data for their calculation and reproducibility.

Regarding 2) An evaluation of outliers as well as four additional examples from the same data set were included in the new version of the Technical Note. Additionally, a new figure (Figure. 2) was included showing Boxplots of end-members projected in the three-dimensional mixing space as a basis for clarity and a better understanding of the example calculations.

Please find at the end of this document the description of the examples and tables with input information (Table 1 to Table 4) and results (Table 5 to Table 8) that have been included and discussed in the Technical Note.

**Major Comments:**

**The authors claim the robustness of their method (P.2, L.34), but do not discuss this consideration in comparison for example, to the commonly applied Gaussian error propagation approach. I think it would be great to evaluate and discuss this in the manuscript to add value to the usefulness of the presented methodology. This could also help differentiate this work from the work of (Correa et al., 2019).**

R: We agree with the Reviewer that an exhaustive comparison of different methods should be attempted at some point, but after careful consideration we did not follow this suggestion here due to the length of this technical note. We on purpose used this format and not a full research paper to present our novel methodology in more mathematical detail than usual, step-by-step with an example application and we also provide the codes. This uncertainty assessment method was not presented in Correa et al. (2019), only parts of the dataset.

**System of equations and resolution: please make sure to define clearly all of the notation in the set of equations to facilitate the readability of their resolution throughout the paper. All the considerations within the resolution of the system must also be clearly stated.**

R: We appreciate this suggestion and have extended and updated the manuscript to clarify the notation and also to include more details in the descriptions of equations and variables to improve the readability of the technical note.

**Specific comments:**
**Page 1, Lines 12-13: this sentence is incomplete. Please correct.**
R: We have edited the phrase. It now reads: "[…], Bayesian approaches to estimate such source uncertainty only exist only sound methods for two and three sources."

**P.1, L15: "dataset".**

R: In this context the word "set" refers to the set of equations used to calculate the uncertainty of the source's contributions to a mixture, not to the data set, therefore we have omitted this change.

**P.2, L.7: Replace "novel" by "the availability of".**

R: This has been corrected to "the availability of"

**P.2, L.12: Delete "novel".**

R: The word "novel" has been deleted.

**P.2, footnote: I think "M" refers to the mixture, not to a source. Correct if necessary.**

R: This has been corrected to "mixture"

**P.7, L.4: n is approximately 30 for each source, or among all sources**

R: We have edited the phrase to clarify this point. It now reads: "[…] and spring water (SW) (n ~ 30, for each end-member) were collected".

**P.7, L.21 and L.23: I think you refer to streamflow (or mixture, M), and not to spring water (SW). Correct if necessary.**

R: We have edited the phrase to correct this error. "SW" was replaced by "M".

**Table 2: report $\lambda$ values**
R: By $\lambda$ we assumed that the Referee refers to degrees of freedom ($\gamma$). These values are reported in the Table 5.

We have generated 6 examples to analyze the sensitivity of the uncertainty calculation to the source sample size, the artificial inclusion of outliers (upper and lower extremes) and the increased standard deviations of the sources datasets. The first example considers 50% of the samples from the initial data sets of sources (Table 1). The median, standard deviation and sample size are input data (Table 2) to calculate the uncertainty ranges (Table 6). The second considers the remaining 50% of samples and was similarly executed (Table 2). In the third example, outliers were artificially included at the positive end of data sets from each source at each coordinate, respectively. The outliers consisted of twice the maximum positive value of the observed data (Table 3). Using the same criteria, the negative extremes were included in the fourth example (Table 3). In order to take into account the effect of sources with dispersed data clouds, the increase of the standard deviation was considered in two cases, the first, in the example five, increasing three times the value of the standard deviation of the initial data set (Table 4) and finally, increasing the standard deviation five times in the sixth example (Table 4).

[revised manuscript text omitted]

---

## Author Comment (AC3) · 16 Sep 2019

Response to comments from the Referee 2:

Major comments: The main problem with this technical note is the lack of definitions and explanations of variables and notations. Albeit I understand that a technical note is supposed to be short, if the authors want this work to be utilized by other geoscientists and hydrologists, they need to improve the readability of the technical note.

R: We appreciate the Reviewer's comment in line with another Reviewer's evaluation and have followed his/her suggestions throughout the document. We now included more details on descriptions of equations, variables and notations to improve the readability of the technical note.

[Figure]

If I understand correctly, this technical note details the methodology used by Correa et al. (2019). It would be great if the authors could be more upfront about this in the technical note.

R: We are grateful for the comment and we agree, in the new version of the Technical Note it is clearly specified that the methodology developed here is the one applied in Correa et al. (2019b). It now reads: "We illustrate this application on the study case published in Correa et al. (2019b), where the authors presented the uncertainty analysis of sources contributions results [. . .]".

The Matlab script is relatively hard to follow and understand, due to insufficient comments and documentation: Naming of Matlab Files: Matlab filenames starting with a number cannot be run by Matlab. Please rename in order to avoid confusion for unexperienced Matlab users. Name of csv file is different in repository than as used in matlab script (5_data.csv rather than data.csv). Please make consistent. The Matlab Script is not very well documented. Neither the script directly, nor the readme pdf explain the actual inputs (only some cryptic A/B/C/D/M without explanations; one has to refer to Table 1 to understand the setup of the file) nor the outputs. Consequently, the script cannot be used without reading the manuscript in detail. It would furthermore be helpful to reference the code lines to the equations of the manuscript and state in the script what each command is doing. Please also reference the manuscript in the readme.pdf and the main script. Specific Comments:

R: We are very grateful that the reviewer highlights the lack of readability in the MatLab script and its documentation. We intended to use the script along with the technical note where the definitions of terms are stated, however, we have included a detailed description in the documentation of the script. Additionally, we have maintained consistency between the repository and the code. The number of equations in the manuscript have been included in the code (method.m) for reference, along with a description of what each equation does. The manuscript (under review) will be referenced in the readme file. Please find as supplement of this document, the updated readme and

method files.

Specific Comments: p.1 line 13: the grammar of this sentence is wrong/the sentence is incomplete.

R: We have edited the phrase. It now reads: "[. . .], Bayesian approaches to estimate such source uncertainty only exist only sound methods for two and three sources."

p.1 lines 15-17: There is no connection between the two sentences, in spite of the "however". Sentence 1 talks about large tracer sets from four water sources (but says nothing about the number of tracers), Sentence 2 says the approach can be generalized to any number of tracers. Please make this clearer.

R: We have updated this section as suggested: "We illustrate the method to compute individual uncertainties in the calculation of source contributions to a mixture, particularly with an example from hydrology, where a 14-tracer set from water sources and streamflow from a tropical, high-elevation catchment were used. Moreover, this method has the potential to be generalized to any number of tracers in a wide range of disciplines."

p.5 line 1-2: the reference of Taylor, 1982 should probably follow directly after "Taylor series approximation".

R: We have edited the phrase. It now reads: "[. . .], the first-order Taylor series approximation (Taylor, 1982) for the variance [. . .]."

p.7 line 2: the specification of n=270 is no useful without also specifying the number of different streams sampled.

R: We thank the reviewer for pointing out a typographical error, the correct number of samples is 257 and the number of streams sampled is 5, the phrase was updated: "Streamwater samples from 5 nested streams were collected weekly from March 2013 to April 2014 (n=257) and at a higher frequency during experimental campaigns."

References: Correa et al, 2018, SciTotEnv should actually be Correa et al. 2019.

R: We updated the reference: Correa, A., Breuer, L., Crespo, P., Célleri, R., Feyen, J., Birkel, C., Silva, C. and Windhorst, D.: Spatially distributed hydro-chemical data with temporally high-resolution is needed to adequately assess the hydrological functioning of headwater catchments, Science of The Total Environment, 651, 1613–1626, doi:10.1016/j.scitotenv.2018.09.189, 2019b. Table 1: Footnote: "three- axes". I believe this should be three-dimensional space. R: We have edited the footnote. It now reads: "Coordinates of end-members and stream (mixture) medians in three-dimensional space (U1, U2 and U3). n represents the sample size"

Table 2: Some of the values provided here do not match up with those calculated by the MATLAB Code. Please verify.

R: We have verified the result from the MatLab code, and the values presented in Table 2 are correct. However, in the Zenodo platform, the data.csv preview shows the data rounded to a decimal and a missing column (Stvd U2), it is necessary to download the.csv file to get the complete data. Updating the code, we will improve the visualization of the data.

Please also note the supplement to this comment:
https://www.hydrol-earth-syst-sci-discuss.net/hess-2019-197/hess-2019-197-AC3-supplement.pdf

**Supplement:**

**Response to comments from the Referee 2:**

**Major comments:**
**The main problem with this technical note is the lack of definitions and explanations of variables and notations. Albeit I understand that a technical note is supposed to be short, if the authors want this work to be utilized by other geoscientists and hydrologists, they need to improve the readability of the technical note.**

R: We appreciate the Reviewer's comment in line with another Reviewer's evaluation and have followed his/her suggestions throughout the document. We now included more details on descriptions of equations, variables and notations to improve the readability of the technical note.

**If I understand correctly, this technical note details the methodology used by Correa et al. (2019). It would be great if the authors could be more upfront about this in the technical note.**

R: We are grateful for the comment and we agree, in the new version of the Technical Note it is clearly specified that the methodology developed here is the one applied in Correa et al. (2019b).
It now reads: "We illustrate this application on the study case published in Correa et al. (2019b), where the authors presented the uncertainty analysis of sources contributions results […]".

**The Matlab script is relatively hard to follow and understand, due to insufficient comments and documentation:**
**Naming of Matlab Files: Matlab filenames starting with a number cannot be run by Matlab. Please rename in order to avoid confusion for unexperienced Matlab users.**
**Name of csv file is different in repository than as used in matlab script (5_data.csv rather than data.csv). Please make consistent.**
**The Matlab Script is not very well documented.**
**Neither the script directly, nor the readme pdf explain the actual inputs (only some cryptic A/B/C/D/M without explanations; one has to refer to Table 1 to understand the setup of the file) nor the outputs.**
**Consequently, the script cannot be used without reading the manuscript in detail.**
**It would furthermore be helpful to reference the code lines to the equations of the manuscript and state in the script what each command is doing.**
**Please also reference the manuscript in the readme.pdf and the main script.**
**Specific Comments:**

R: We are very grateful that the reviewer highlights the lack of readability in the MatLab script and its documentation. We intended to use the script along with the technical note where the definitions of terms are stated, however, we have included a detailed description in the documentation of the script. Additionally, we have maintained consistency between the repository and the code.
The number of equations in the manuscript have been included in the code (method.m) for reference, along with a description of what each equation does.
The manuscript (under review) will be referenced in the readme file.
Please find at the end of this document, the updated readme and method files.

**Specific Comments:**
**p.1 line 13: the grammar of this sentence is wrong/the sentence is incomplete.**

R: We have edited the phrase. It now reads: "[…], Bayesian approaches to estimate such source uncertainty only exist only sound methods for two and three sources."

**p.1 lines 15-17: There is no connection between the two sentences, in spite of the "however". Sentence 1 talks about large tracer sets from four water sources (but says nothing about the number of tracers), Sentence 2 says the approach can be generalized to any number of tracers. Please make this clearer.**

R: We have updated this section as suggested: "We illustrate the method to compute individual uncertainties in the calculation of source contributions to a mixture, particularly with an example from hydrology, where a 14-tracer set from water sources and streamflow from a tropical, high-elevation catchment were used. Moreover, this method has the potential to be generalized to any number of tracers in a wide range of disciplines."

**p.5 line 1-2: the reference of Taylor, 1982 should probably follow directly after "Taylor series approximation".**

R: We have edited the phrase. It now reads: "[…], the first-order Taylor series approximation (Taylor, 1982) for the variance […]."

**p.7 line 2: the specification of n=270 is no useful without also specifying the number of different streams sampled.**

R: We thank the reviewer for pointing out a typographical error, the correct number of samples is 257 and the number of streams sampled is 5, the phrase was updated: "Streamwater samples from 5 nested streams were collected weekly from March 2013 to April 2014 (n=257) and at a higher frequency during experimental campaigns."

**References: Correa et al, 2018, SciTotEnv should actually be Correa et al. 2019.**

R: We updated the reference: Correa, A., Breuer, L., Crespo, P., Célleri, R., Feyen, J., Birkel, C., Silva, C. and Windhorst, D.: Spatially distributed hydro-chemical data with temporally high-resolution is needed to adequately assess the hydrological functioning of headwater catchments, Science of The Total Environment, 651, 1613–1626, doi:10.1016/j.scitotenv.2018.09.189, 2019b.

**Table 1: Footnote: "three- axes". I believe this should be three-dimensional space.**
R: We have edited the footnote. It now reads: "Coordinates of end-members and stream (mixture) medians in three-dimensional space (U1, U2 and U3). n represents the sample size"

**Table 2: Some of the values provided here do not match up with those calculated by the MATLAB Code. Please verify.**

R: We have verified the result from the MatLab code, and the values presented in table 2 are correct. However, in the Zenodo platform, the data.csv preview shows the data rounded to a decimal and a missing column (Stvd U2), it is necessary to download the.csv file to get the complete data.
Updating the code, we will improve the visualization of the data.

Readme file

These codes estimate the uncertainty of individual end-member (source) contributions to streams (mixture) based on a multi-tracer set in a three-dimensional space.
The method.m code shows step-by-step calculations of partial derivatives, degrees of freedom, t-Student and confidence interval limits for each source fraction.
method.m uses the functions Yx.m and dYzdyx.m for its execution.
A, B, C and D represent the set of sources and M the mixture.
Please refer to Correa et al., (2019) in Correa, A., Ochoa-Tocachi, D. and Birkel, C.: Technical note: Uncertainty in multi-source partitioning using large tracer data sets, Hydrology and Earth System Sciences Discussions, 1–14, doi:https://doi.org/10.5194/hess-2019-197, 2019 for a very detailed description of the used notation, equations and variables for this example.
The equation numbers used in the code method.m refer to the corresponding ones in Correa et al., (2019).

Instructions:
Enter data for the median of end-members and mixture, their standard deviations and sample size, all projected in
the three-dimensional PCA-space (U-space in the referred Technical Note). The file must be named data.csv and follows an order similar to the one presented in this example:

|   | Median in U1 | Median in U2 | Median in U3 | Stvd in U1 | Stvd in U2 | Stvd in U3 | No. samples | No. samples | No. samples |
|---|---|---|---|---|---|---|---|---|---|
| A | 26.25 | 7.29 | 7.00 | 0.46 | 0.36 | 0.4 | 25 | 25 | 25 |
| B | 0.23 | 5.47 | 1.98 | 0.85 | 1.29 | 0.69 | 33 | 33 | 33 |
| C | -2.24 | -3.93 | 3.74 | 0.55 | 0.58 | 0.45 | 37 | 37 | 37 |
| D | -5.38 | -6.1 | -4.85 | 0.27 | 0.56 | 0.15 | 36 | 36 | 36 |
| M | -0.61 | -1.04 | 0.94 | 2.07 | 1.1 | 0.66 | 257 | 257 | 257 |

Please do not include rows and column names in the data.csv file, here is used for merely visual purposes of the example.

Run the method.m code.
It is recommended to keep all files in a common directory (method.m; dYzdyx.m; Yx.m and data.csv)
After executing the code, the fractions of the contribution of each end member to the mixture are calculated, as well as the
degrees of freedom and upper and lower limits of uncertainties associated with their contribution.

Note: the order of the inputs (end members) is reflected in the outputs (fractions of contribution and uncertainties).

**method file**

```matlab
clear, clc;
format short;
%% Data
%% DATA reads the data.csv file with the input information
DATA = csvread('data.csv');
% Ms
Z = [DATA(:,1:3), ones(5,1)];
Z = transpose(Z);
dataM = Z(:,end);
Z = Z(:, 1:end-1);
% Variance
vZ = transpose (DATA(:,4:6)).^(2);
% Sample QTY
nZ = transpose (DATA(:,7:9));
%% Compute the fractions of sources (A, B, C, D) contribution to the mixture (M) (Eq. 2 and implicitly from Eq. 3)
% x in {A,B,C,D,M} = {1,2,3,4,5}
% y in {delta,lambda,phi} = {1,2,3}
f = zeros(4,1);
DYx = @(y,x) Yx(y,x, Z, dataM);
Num = (DYx(1,5)-DYx(2,5))*(DYx(3,3)-DYx(1,3)) - (DYx(1,3)-DYx(2,3))*(DYx(3,5)-DYx(1,5));
Den = (DYx(1,1)-DYx(2,1))*(DYx(3,3)-DYx(1,3)) - (DYx(1,3)-DYx(2,3))*(DYx(3,1)-DYx(1,1));
f(1) =  Num/Den;
f(3) = ((DYx(1,5)-DYx(2,5))-(DYx(1,1)-DYx(2,1))*f(1))/(DYx(1,3)-DYx(2,3));
f(2) = DYx(1,5) - (DYx(1,3)*f(3) + DYx(1,1)*f(1));
f(4) = 1 - (f(1) + f(2) + f(3));
%% Compute the partial derivatives for fA, fC, fB and fD (Eq. 4 and implicitly from Eq. 5 to Eq. 8)
% x in {A, B, C, D, M} = {1,2,3,4,5}
% Y,y in {delta,lambda,phi} = {1,2,3}
%% dfAdyx, presents the partial derivative for fA (Eq. 9)
DdYzdyx = @(Y,z,y,x) dYzdyx(Y,z,y,x,Z,dataM);
dfAdyx = @(y,x) Den^(-2)*( ...
    ((DYx(2,3)-DYx(1,3))*(DdYzdyx(3,5,y,x) - DdYzdyx(1,5,y,x)) + ...
     (DYx(3,5)-DYx(1,5))*(DdYzdyx(2,3,y,x)-DdYzdyx(1,3,y,x)) - ...
     (DYx(3,3)-DYx(1,3))*(DdYzdyx(2,5,y,x)-DdYzdyx(1,5,y,x)) - ...
     (DYx(2,5)-DYx(1,5))*(DdYzdyx(3,3,y,x)-DdYzdyx(1,3,y,x)))*Den - ...
    ((DYx(2,3)-DYx(1,3))*(DdYzdyx(3,1,y,x)-DdYzdyx(1,1,y,x)) + ...
     (DYx(3,1)-DYx(1,1))*(DdYzdyx(2,3,y,x)-DdYzdyx(1,3,y,x)) - ...
     (DYx(3,3)-DYx(1,3))*(DdYzdyx(2,1,y,x)-DdYzdyx(1,1,y,x)) - ...
     (DYx(2,1)-DYx(1,1))*(DdYzdyx(3,3,y,x)-DdYzdyx(1,3,y,x)))*Num);
 dfCdyx = @(y,x) ((DYx(1,3)-DYx(2,3))^(-2))*( ...
    ((DdYzdyx(1,5,y,x)-DdYzdyx(2,5,y,x)) - ...
     (DdYzdyx(1,1,y,x)-DdYzdyx(2,1,y,x))*f(1) - ...
     (DYx(1,1)-DYx(2,1))*dfAdyx(y,x))*(DYx(1,3)-DYx(2,3)) - ...
     (DdYzdyx(1,3,y,x)-DdYzdyx(2,3,y,x))*( ...
     (DYx(1,5)-DYx(2,5))-(DYx(1,1)-DYx(2,1))*f(1)));
dfBdyx = @(y,x) DdYzdyx(1,5,y,x) - DdYzdyx(1,3,y,x)*f(3) - ...
    DYx(1,3)*dfCdyx(y,x) - DdYzdyx(1,1,y,x)*f(1) - DYx(1,1)*dfAdyx(y,x);
dfDdyx = @(y,x) -dfCdyx(y,x)-dfBdyx(y,x)-dfAdyx(y,x);
%% Compute the variancefor each end-member fraction, fA, fB, fC and fD respectively (Eq. 10 and Eq. 13)
% x in {A,B,C,D,M} = {1,2,3,4,5}
% y in {delta,lambda,phi} = {1,2,3}
v = zeros(4,1);
for x = 1:5
    for y = 1:3
        v(1) = v(1) + (dfAdyx(y,x)^2)*vZ(y,x);
        v(2) = v(2) + (dfBdyx(y,x)^2)*vZ(y,x);
```

```matlab
            v(3) = v(3) + (dfCdyx(y,x)^2)*vZ(y,x);
            v(4) = v(4) + (dfDdyx(y,x)^2)*vZ(y,x);
        end
    end
%% Satterthwaite degrees of freedom for each end-member fraction (Eq. 12 and Eq. 14).
% x in {A,B,C,D,M} = {1,2,3,4,5}
% y in {delta,lambda,phi} = {1,2,3}
g = zeros(4,1);
for x = 1:5
    for y = 1:3
        g(1) = g(1) + (((dfAdyx(y,x)^2)*vZ(y,x))^2)/(nZ(y,x)-1);
        g(2) = g(2) + (((dfBdyx(y,x)^2)*vZ(y,x))^2)/(nZ(y,x)-1);
        g(3) = g(3) + (((dfCdyx(y,x)^2)*vZ(y,x))^2)/(nZ(y,x)-1);
        g(4) = g(4) + (((dfDdyx(y,x)^2)*vZ(y,x))^2)/(nZ(y,x)-1);
    end
end
g = (v.^2)./g;
%% Student's t value(two-tailed) to compute 95% confidence intervals (Walpole et al., 2017)
t = tinv(0.95,g);
%% Compute the upper and lower confidence interval limits for each end-member fraction (Eq. 15)
ulim = min(1,f + t.*sqrt(v));
llim = max(0,f - t.*sqrt(v));
%% Present results:
%% f, fractions of sources (A,B,C,D) contribution to the mixture (M),
%% g, degrees of freedom for each end-member fraction
%% ulim and llim, upper and lower confidence interval limits for each end-member fraction
f
g
ulim
llim
```

---

## Author Comment (AC4) · 16 Sep 2019

[revised manuscript text omitted]
 3). In order to take into account the effect of sources with dispersed data clouds, the increase of the standard deviation was considered in two cases, the first, in the example five, increasing three times the value of the standard deviation of the initial data set (Table 4) and finally, increasing the standard deviation five times in the sixth example (Table 4).

The results of this analysis are presented in Tables 6-8. In examples 1 and 2 the sample size reduction from 24 to 12 and 13 samples respectively (Table 6), had a minimal effect (less than 3%) on the calculation of the uncertainty ranges compared to the original complete set (Table 1). The fractions of source contributions do not experienced changes. The inclusion of outliers affected the values of the medians at levels of the second decimal (Table 3) in relation to the median of the initial data (Table 2), however the standard deviations increased in a range of 1.2 to 2.5 times the original value for AN and HS, in greater degree for RF (2.5 to 10.5) and drastically for SW (4 to 20 times wider). These variations were reflected in the results of the calculation of uncertainties where the limits were extended for all existing cases from 1% to 12% (Table 6) in relation with the presented in the Table 5. The location of the SW (source with high solutes concentrations (Correa et al., 2019b) in the three-dimensional space, turned it highly sensitive to the outlier inclusion presented in the examples. 
[revised manuscript text omitted]

---

## Author Response (AR2)

**Response to comments from the Referee 2:**

**Report #1**

**Anonymous Referee #2**

**In my previous comments on the manuscript I mainly remarked on the lack of explanations in the manuscript and the lack of documentation in the MATLAB code. I am satisfied with the changes made by the authors to the manuscript during the revision. The technical note is substantially easier to read now, and I recommend publication. The changes made to the MATLAB script and documentation are also very helpful for an improved understanding.**

R: We are very thankful to the Referee's useful remarks, which greatly helped to improve our Technical Note. We followed his/her final suggestions throughout the document.

**In the following are a few minor details that need improvement in the manuscript before publication:**

**Section 2: "Let C, represents the set of sources" – should be "let C represent the set of sources" instead.**

R: This has been corrected to "let C represent the set of sources"

**"where $f_A, f_B, f_C$ and $f_D$ represent the contribution fraction of sources A, B, C and D respectively to the mixture M and Eq. (1) has solution1for $f_A, f_B, f_C, f_D$>0, they take the following form" – this sentence should be split in two, otherwise it is unclear and grammatically incorrect.**

R: We have edited the paragraph to clarify this point. It now reads: "[…] and $z$ to the set of A, M and C. Furthermore, $f_A$, $f_B$, $f_C$ and $f_D$ represent the contribution fraction of sources A, B, C and D respectively to the mixture M.

If the system is composed of Eq. (1)

$$\begin{cases} \overline{\delta}_A f_A & + & \overline{\delta}_B f_B & + & \overline{\delta}_C f_C & + & \overline{\delta}_D f_D & = & \overline{\delta}_M \\ \overline{\lambda}_A f_A & + & \overline{\lambda}_B f_B & + & \overline{\lambda}_C f_C & + & \overline{\lambda}_D f_D & = & \overline{\lambda}_M \\ \overline{\phi}_A f_A & + & \overline{\phi}_B f_B & + & \overline{\phi}_C f_C & + & \overline{\phi}_D f_D & = & \overline{\phi}_M \\ f_A & + & f_B & + & f_C & + & f_D & = & 1 \end{cases} \qquad \text{Eq.(1)}$$

and has solution[1] for $f_A, f_B, f_C, f_D > 0$, the contribution fractions take the following form:

**4 "our methodology developed" – should be "our methodology was developed"**

R: This has been corrected to "Our methodology was developed".

**Response to comments from the Referee 1:**

**Report #2**

**Anonymous Referee #1**

**Major comment:**

**- I understand the decision of the authors to present this work in the format of a technical note. Yet, their statements along the manuscript should be aligned to the proposed method and the presented analyses only. Thus, even when I agree that their methods is very valuable for the proposed purpose, a formal evaluation of the robustness of the method in comparison to other methodologies is not presented. In the new version of the manuscript the authors claim the robustness of their method based on the analysis of the effect of different data inputs on the resulting source uncertainties (P.10 , L.172-174). However, this analysis does not provide more information than the sensitivity of the estimated uncertainties to input data with different conditions. Thus, I strongly suggest the authors to avoid any misleading conclusion about the supposed robustness of their method throughout the manuscript, particularly in section 4 and P.2, L.53.**

We acknowledge the Referee's comment, which helped us to follow a more precise line. In fact, in this Technical Note, no comparative analysis with other methodologies has been performed and therefore we have carefully reviewed the entire document, to avoid the misinterpretation of the word robust. Besides, we focus on a better description of the methodological development and application examples.

**Minor comments:**

**P.1,L.12-13: This sentence is still difficult to understand. Do you mean "However, the source contributions may be uncertain and to date only Bayesian approaches to estimate the uncertainty of two and three sources exist." Or something along those lines. Revise sentence for clarity.**

R: We have edited the phrase to clarify this point. It now reads: "[…] the source contributions may be uncertain and apart from Bayesian approaches, to date there are only solid methods to estimate such uncertainties for two and three sources".

**P1,L13: replace "expand this methods developing an" by "introduce an alternative".**

R: This has been modified for "introduce an alternative".

**P1,L16: delete "particularly".**

R: The word "particularly" has been deleted.

**P1,L18: delete "were used"**

R: The words "were used" have been deleted.

**P2,L30-31: Avoid double parenthesis.**

R: Done

**P2,L33: "tracer mass balance"**

R: The word "tracer" has been included.

**P2,L34: sources and dynamics of what? Specify for clarity.**

R: We have edited the phrase to clarify this point. It now reads: "[…] mixing models based on tracer mass balance equations are widely-applied to identify the dominant sources of a mixture and their contribution dynamics".

**P2,L39: "mixing space"**

R: The word "mixing" has been included.

**P2,L47: "and their individual uncertainty"**

R: This has been modified for "and their individual uncertainty".

**P2,L53: replace "a novel and robust" by "an alternative"**

R: This has been corrected to "an alternative".

**P2.,L54: perhaps good be good to give examples of what sources and mixture refer to. For instance "end members or sources (e.g., precipitation, soil water, snowmelt) to a mixture (e.g., streamflow)"**

R: We have included this suggestion.

**P2,L55-60: These 2 long sentences could be split at least into 4-5 shorter ones to make it easier to read. Also, I think that mentioning the "application of a final equation" is not the best way to mention that the methodology has already been a applied without a formal description of the method. Please re-phrase for clarity.**

R: We have split the sentences and reworded the paragraph for clarity. It now reads:

"We illustrate this application using a multi-tracer data set from Correa et al. (2019b), in a three-dimensional space defined by a Principal Component Analysis. In  Correa et al. (2019b), the authors calculated the uncertainties but without disclosing any details in the calculation and methodology used. The main objective of this Technical Note is therefore to explicitly describe the mathematical development that allows the calculation of partial derivatives, degrees of freedom and confidence interval limits for each source fraction contribution. Moreover, to provide the code and several examples for their calculation and reproducibility".

**P2,L58: Noting was mentioned about the method in the rest of the introduction, so this sentence comes as a complete surprise. I suggest mentioning something about the proposed Taylor series approximation around P1,L54 so here you relate it directly to "the calculation of partial derivatives, degrees of freedom and confidence interval limits".**

R: We have reworded the paragraphs for clarity. It now reads:

Around P1,L54 "[…] we propose an alternative methodology based on the first-order Taylor series approximation to estimate the uncertainty […]".

Around P1,L58 "[…] allows the calculation of partial derivatives, degrees of freedom and confidence interval limits for each source fraction contribution […]".

**P3,L62-70: It would be helpful to specify if each of the variables correspond to vectors and matrices and what is the specific data related to these variables.**

R: We have included what data are necessary for the analysis related to the sources and the mixture and indicated that further details are presented in section 3.2. The way to use the data (as matrix) in the script is detailed in section 3.3, where the practical exercises are applied.

The paragraph now reads:

"The data required for this analysis are the median and standard deviations ($\sigma$) of each of the sources (A, B, C and D) and the mixture M, projected and expressed in the coordinates of the three-dimensional PCA space. In addition, the sample size (n) of each source is required. Details of the application are presented in section 3.2."

**P3,L69-70: unclear, please split into 2 sentences for clarity. Also, I do not see the need to use a footnote. Foot note 1 could easily be included in this short paragraph.**

R: We have edited the paragraph to clarify this point. It now reads: "[…] and $z$ to the set of A, M and C. Furthermore, $f_A, f_B, f_C$ and $f_D$ represent the contribution fraction of sources A, B, C and D respectively to the mixture M.

If the system is composed of Eq. (1)

$$
\begin{cases}
\overline{\delta}_A f_A & + & \overline{\delta}_B f_B & + & \overline{\delta}_C f_C & + & \overline{\delta}_D f_D & = & \overline{\delta}_M \\
\overline{\lambda}_A f_A & + & \overline{\lambda}_B f_B & + & \overline{\lambda}_C f_C & + & \overline{\lambda}_D f_D & = & \overline{\lambda}_M \\
\overline{\phi}_A f_A & + & \overline{\phi}_B f_B & + & \overline{\phi}_C f_C & + & \overline{\phi}_D f_D & = & \overline{\phi}_M \\
f_A & + & f_B & + & f_C & + & f_D & = & 1
\end{cases}
\qquad \text{Eq.(1)}
$$

and has solution[1] for $f_A, f_B, f_C, f_D > 0$, the contribution fractions take the following form:

However, we preferred to keep the footnote to avoid including a new equation and further complicating this section.

**P5,L76: add symbol of variance**

R: The symbol of variance $(\sigma^2)$ has been included.

**P5,L76: define cA**

The definition of "cA" has been included. It now reads: "[…] where $c_A$ is a scale constant that relates $f_{Ay_x}$ with the respective derivative. It means that $f_A$ with respect to $y_x$ can be a scalar multiple of the derivative value.

**P5,Eq. 11: define n**

R: It was previously defined in the line 73

**P6,L85: crossreference Eq6**

R: We have cross-referenced, however the correct equation is Eq.(10).

**P7,L99: the IUSS reference is the general classification of soils, not the proportions of each of them at your study site as stated. Suggest deleting this reference and use one specific for the study area.**

R: We have updated the reference to: (Quichimbo et al., 2012)

**P7,L107-108: suggest moving these results from the cited references to L.115, so it is clear what the end members of the system are and easier to relate them to the rest of this section.**

R: We have updated this section as suggested. It now reads: "[…] data from waters sources RF, AN, HS and SW, were projected into a three-dimensional space (Correa et al., 2019b) and presented in Figure 1 and Table 1".

**P7,L120-124 & P8,L125: This is basically a repetition of the methods section. Why not simply mention that A,B,C, and D now are represented by end member RF,AN,HS, and SW in the corresponding equations to shorten the text?**

R: We agree, we have eliminated this redundant paragraph and highlighted the correspondence between the end-members used in the example and the terminology in the equations.

**P8,L125: Suggest to keep using the same notation than in the methods across the whole manuscript (i.e., A,B,C,D instead of EM1,E2,EM3,EM4). After all, that is the same notation used in tables 1-4. However, whatever your decision, everything needs to be consistent, i.e., correct in tables 5-8.**

R: Yes, we agree, we have maintained consistency using A,B,C and D throughout the document.

**P8,L127-128: "U1, U2 and U3 represent the principal components PC1,PC2 AND PC3, respectively"**

R: This section has been updated. It now reads: "[…] U1, U2 and U3 represent the principal components PC1,PC2 AND PC3, respectively"

**P9,L129: "… procedure was applied to all…"**

R: This has been modified for "A similar procedure was applied to all end-members".

**P9,L140-155: perhaps would be best to include this description using an additional section to the paper eg.: 3. Sensitivity of source uncertainty to input data. Then, a subsection with the same suggested name could be added to section 3. Application to describe the results of this analysis. For now, this part appears as a surprise to the reader.**

R: We greatly appreciate this comment, we agree that the creation of a new section (3.3) will facilitate the readability of the document.

**P9,L143-144: delete, repeated in the next sentence.**

R: The paragraph has been deleted.

**P9,L145: report how the 50% of data in set 1 was selected.**

R: The data set was divided in chronological order of sample collection, the samples gathered in the first half of the monitoring period (50%) were considered for example 1 and the remainder (50%) for example 2.

The reason was reported, and it now reads: "The first example considers 50% of the samples (collected in the first half of the monitoring period) from each source".

**P9,L148: the second "example"**

R: The word "example" has been included.

**P9,L160-162: rewrite sentence for clarity.**

R: We have edited the paragraph to clarify this point. It now reads: "These variations were reflected in the widening (1% to 12%) of uncertainty bands for all existing cases (Table 7) in comparison with those calculated from the original data set (Table 5)".

**P10,L172: delete "been" and consider my major comment with regards to the "robustness" of the method.**

R: The word "example" has been deleted and your major comment fully considered.

**P10,L79: "… a larger number of source contributions (>3) and the…"**

R: This has been modified for "a larger number of source contributions (>3) and the".

**Tables 5-8: to keep consistency throughout the manuscript I suggest you use the notation A,B,C,D instead of the EMx notation.**

R: Yes, we agree, we have maintained consistency using A,B,C and D throughout the document.

[revised manuscript text omitted]